# Health Literacy Associations with Periodontal Disease among Slovak Adults

**DOI:** 10.3390/ijerph17062152

**Published:** 2020-03-24

**Authors:** Silvia Timková, Tatiana Klamárová, Eva Kovaľová, Bohuslav Novák, Peter Kolarčik, Andrea Madarasová Gecková

**Affiliations:** 11st Department of Stomatology, Faculty of Medicine, P.J. Šafarik University, 04011 Košice, Slovakia; silvia.timkova@upjs.sk; 2Department of Dental Hygiene, Faculty of Health Care, University of Prešov, 08001 Prešov, Slovakia; tklamarova@gmail.com (T.K.); kovalova@nextra.sk (E.K.); 3Department of Stomatology and Maxilofacial Surgery, Faculty of Medicine, Comenius University, 81250 Bratislava, Slovakia; bohusnovak@gmail.com; 4Department of Health Psychology and Research Methodology, Faculty of Medicine, P.J. Šafárik University, 04011 Košice, Slovakia; andrea.geckova@upjs.sk; 5Olomouc University Social Health Institute—OUSHI, Palacký University, 77111 Olomouc, Czech Republic

**Keywords:** health literacy, oral health, dental hygiene, Community Periodontal Index of Treatment Needs (CPITN), Health Literacy Questionnaire (HLQ)

## Abstract

Periodontal disease is inflammation of the gums and without good oral hygiene, it can progress to periodontitis. Oral hygiene might be related to a patient’s health literacy (HL), defined as ability to gain access, understand, and use information to promote and maintain good health. The aim of our study is to examine the associations of HL with indicators of periodontal disease. A cross-sectional study on 1117 adults (36.2% males; mean age = 36.4, SD = 14.2) attending dental hygiene treatment was conducted. Data on demographics, socioeconomic status, and nine domains of HL (Health Literacy Questionnaire, HLQ) were collected by questionnaire, and Community Periodontal Index of Treatment Needs (CPITN) was established by the dental hygienist. Data were analysed using t-tests and logistic regression. Respondents with periodontal disease (N = 152) had statistically significantly lower levels of HL in seven out of nine HLQ domains compared to intact patients (N = 818) (t from 3.03 to 4.75, *p* < 0.01). Association of higher HL in seven domains with lower chance of diagnosed periodontal disease remain significant even after adjustment for age, gender and educational attainment (adjusted ORs 0.55–0.67, *p* < 0.05). Our findings confirm that an individual’s lower HL is significantly associated with higher chance of periodontal disease incidence, specifically among Slovak adults attending oral hygiene clinics. HL might be a promising factor in the improvement of oral health in this population, worthy of consideration in intervention and preventive activities.

## 1. Introduction

The term periodontal disease refers to the inflammation of the gums and soft tissue surrounding the teeth in response to bacterial accumulations, or dental plaque, on the teeth [1]. The disease is usually a result of improper oral hygiene subsequently causing a build-up of plaque and calculus on the teeth and interdental space. Periodontal disease has several states or stages. If the first stage of gum inflammation, gingivitis, is not treated or prevented by proper oral hygiene it can lead to periodontitis, which is a more advanced state of gum disease that affects the bone and is more difficult to treat [2]. An individual’s ability to clean his or her teeth properly might be related to the patient’s level of health literacy (HL), defined by World Health Organization [3] as the cognitive and social skills which determine the motivation and ability of individuals to gain access to, understand and use information in ways which promote and maintain good health. Such a relationship has not been studied yet.

The oral cavity is an ideal environment where many different types of microorganisms proliferate and create the oral microbiome. Whether gum tissue is healthy depends on the balance in the composition of each species. Conversely, changes in the balance between different bacterial species that live in the human body contribute to the pathogenesis of some diseases [4]. A change in the composition of the subgingival microbiome is one of the most common causes of periodontal disease [5,6]. Periodontitis is a microbial infection that is manifested by inflammation of the gingival tissue. Bacteria, especially gram-negative species, damage the pendulum apparatus of the teeth, forming a subgingival biofilm [7]. Our immune system reacts to its presence and pro-inflammatory molecules are released. The extent of damage to the epithelium, gingiva, pendulum apparatus of teeth/gingival connective tissue, cement and bone depends on the type of immune defense response of each individual [5,6]. This suggests that periodontitis may manifest concurrently with other systemic diseases [1]. The relationship between periodontitis and systemic diseases such as cardiovascular disease, diabetes, respiratory disorders, rheumatoid arthritis, cancer and adverse pregnancy outcomes has been increasingly recognized in the last two decades [8]. The impact of oral infection on overall health has been defined by a new area of periodontology known as “periodontal medicine” [8].

There are several indexes used for assessment of periodontal health, such as the periodontal disease index (PDI), the community periodontal index of treatment needs (CPITN) and the periodontal screening and recording (PSR) [9], and their modifications, such as the basic periodontal examination (BPE) [10]. CPITN was developed by WHO to provide a valid screening and epidemiological tool for quick assessment of periodontal health in larger samples and easy comparison among populations [11].

Severe periodontal disease, which can cause tooth loss, occurs in 15%–20% of middle-aged adults (35–44 years). Globally, about 30% of people aged 65–74 have no natural teeth [12]. Gingivitis occurs at any age, and more than 90% of the world’s population suffer from gingivitis at some stage. Developed periodontitis is the sixth most common oral disease. At the same time, individual, home-based oral hygiene is not sufficient to achieve optimal oral health [13].

Health literacy in the realm of oral health is a relatively new concept. In health research, it was found that poor health literacy is associated with a wide range of health-related results, including poorer health, poorer use of preventive health care, higher mortality and more hospitalizations [14,15,16]. Although research of oral health and health literacy is not sufficient, there are few studies showing positive association of higher HL with better periodontal health [17], and associations of lower HL with untreated caries, less frequent tooth brushing and irregular flossing [18], fewer visits to the dentist and more frequent emergency room visits for nontraumatic dental conditions [19].

A higher level of knowledge about oral health literacy is important to improve people’s awareness of the presence of oral health complications and dental caries, as well as dissemination of knowledge about the prevention of these diseases to improve and promote individual oral health behavior [20].

Although ill oral health can be avoided by proper oral health care and preventive activities, this problem persists in many countries around the world [21]. The aim of our study is to examine the associations of HL measured with the Health Literacy Questionnaire with Community Periodontal Index of Treatment Needs (CPITN) treated as an indicator of gingivitis among Slovak adults controlling for age, gender and educational attainment of the sample.

## 2. Materials and Methods

### 2.1. Sample and Procedure

A cross-sectional study was conducted in dental hygiene clinics (N = 28) across Slovakia in 2018 (N = 1117 adults; 36.2% of males; mean age = 36.4, SD = 14.2). Data were collected by self-administered questionnaire and they were combined with objective clinical oral health data both collected by a trained dental hygienist (i.e., Community Periodontal Index of Treatment Needs (CPITN)). Participating dental hygiene clinics were recruited from the collaborating dental hygiene clinics (N = 35) where the Department of Dental Hygiene (Faculty of Health Care, University of Prešov) and the First Department of Stomatology (Faculty of Medicine, P.J. Šafarik University) provided training, supervision or consultations for dental hygienists in those clinics. All dental hygiene clinics were private clinics which reflects the situation in Slovakia. Patients attending dental hygiene treatment older than 18 years were asked to participate in the study. Brief explanation of the study rationale and informed consent form were provided by the hygienist before treatment. Patients who agreed to participate filled in the questionnaire after dental hygiene treatment. From the 1479 invited patients 75.5% participated in the study.

The study was approved by the Ethics Committee of the Faculty of Medicine at the P.J. Šafárik University in Kosice and by the Ethics Committee of Louis Pasteur University Hospital in Kosice. Participation in the study was voluntary and anonymous with no explicit incentives provided for participation.

### 2.2. Measures

A self-administered questionnaire was used to acquire subjective data on the respondents. The battery of questions contained various items and scales covering different variables related to respondents’ oral health and health related behaviours. For the purposes of this paper, we used demographics (e.g., gender, age), socioeconomic status (educational attainment) and the health literacy questionnaire (HLQ-SK). The questionnaire battery was in the Slovak language; originally English questionnaires were used in the Slovak version, but it was officially translated and validated. Objective oral health status was evaluated using Community Periodontal Index of Treatment Needs (CPITN).

Socio-demographic variables were categorized subsequently: gender as male (1) or female (2), educational attainment as primary education (1), high school without graduation (2), high school with graduation (3), college/university education (4). Age data was used as continuous variable.

Health literacy was measured using the Slovak version of the health literacy questionnaire (HLQ-SK) [22,23] which consisted of 44 items divided into 9 subscales of health literacy (see Table 1). Translation, adaptation and validation of the questionnaire followed specific translation procedures developed by the HLQ authors. HLQ-SK replicated factor structure of the English HLQ factor structure (satisfactory goodness of fit [χ2WLSMV = 1684.96 (df = 866), *p* < 0.001; CFI = 0.943, TLI = 0.938, RMSEA = 0.051, and WRMR = 1.297] and achieved acceptable internal consistency and component reliability; Cronbach’s alphas and composite reliability coefficients ranged from 0.73 to 0.84 [23]. The original HLQ is divided into two parts which differ in response categories. Part 1 (domains 1–5) has 4 response categories rating the extent of agreement (see Table 1). Part 2 (domains 6–9) has 5 response categories rating the level of difficulty: cannot do or always difficult (1), usually difficult (2), sometimes difficult (3), usually easy (4) and always easy (5). Each domain was scored as the average of the item scores [23]. Higher score indicates a higher level of HL abilities in a particular domain.

Oral health status (presence of periodontal disease) was expressed as a specific standardized index called Community Periodontal Index of Treatment Needs (CPITN) [24]. CPITN was originally designed for describing periodontal treatment needs in populations. It has also been used to describe the prevalence of periodontal conditions and as a screening test to identify patients who need complex or simple treatment [25]. CPITN records the common treatable conditions: periodontal, pockets, gingival inflammation (identified by bleeding on a gentle probing), and dental calculus and other plaque retentive factors. It does not record irreversible changes such as recession or other deviation from periodontal health such as tooth mobility or loss of periodontal attachment. The dentition is divided into six parts (sextants: 17–14, 13–23, 24–27, 37–34, 33–43, 44–47) and each sextant is given a score. The highest score in each sextant is identified after examining all teeth: no need for care (score 0), bleeding gingivae on gentle probing (score 1), presence of calculus and other plaque retentive factors (score 2), and presence of 4 or 5 mm pockets (score 3) or 6 or deeper pockets (score 4). Use of a special 621 WHO periodontal probe (or its equivalent) is recommended [11]. We used CPITN as a dichotomized indicator of healthy teeth (score 0, coded as 0) and teeth with periodontal disease (scores 1–4, coded as 1).

### 2.3. Data Analysis

Firstly, descriptive statistics were used to provide average scores and proportions of respondents in the sample. Further, differences in nine HLQ domains between patients with and without gingival problems were tested using a t-test. The effects of health literacy domains on the presence of gingivitis were analysed using logistic regression and the effect was adjusted for age, gender and educational attainment. Employment status was excluded from the adjusted model because of non-significant association with CPITN at the level of *p* < 0.05. Statistical analyses of the respondents’ data were performed using statistical software package IBM SPSS 23.0 (IBM Corp. Released 2015. IBM SPSS Statistics for Windows, Version 23.0. Armonk, NY, USA.).

## 3. Results

Description of the sample shows that there was a higher prevalence of women in the sample without a statistically significant difference in age. Men and women did not differ in the prevalence of gingivitis. Respondents with gingivitis had significantly higher age compared to respondents without gingivitis (Table 2).

In the Table 3 we present average scores of all HLQ domains separately for the group of patients with gingivitis and for patients with healthy teeth (without signs of gingivitis), respectively. We found that respondents with gingivitis (N = 152) had statistically significantly lower levels of HL in most of HLQ domains (seven out of nine) compared to patients with healthy teeth (N = 818) (Table 3). Further, the results of logistic regression analyses confirmed the association outlined by the t-test and confirmed that such association remained significant even after adjustment for gender, age and educational attainment of the respondents (Table 4). Respondents did not differ significantly in the level of domain “Feeling understood and supported by healthcare providers” (domain one) and domain “Social support for health” (domain four). Lower levels of the other HL domains are then associated with a higher chance of having higher scores of CPITN, which reflects a higher chance of having some form of gingivitis.

## 4. Discussion

Our study aimed to test the associations of HL with the Community Periodontal Index of Treatment Needs (CPITN) treated as an indicator of gingivitis among Slovak adults. We found that respondent’s lower health literacy was significantly associated with a higher chance of gingivitis prevalence, which indicates problematic oral health usually related to improper oral hygiene. We also found that this association was valid after adjustment for age, gender and educational attainment of the respondent. Higher prevalence of gingivitis was seen among older respondents.

Association of lower health literacy with a higher chance of suffering from gingivitis or ill oral health is in accordance with many studies reporting similar associations of patients with low health literacy and their low general health status or having other health issues compared to patients with higher health literacy [14,15,16]. Our results are also consistent with studies on oral health issues and health literacy, even though these types of studies are scarce [17,18,19].

Health literacy, as a characteristic of an individual’s knowledge and skills, is related not just to patients’ processing of health information but also to the application of one’s health. HLQ has nine distinct domains of health literacy in which “Active health management” and “Understand health information well enough to know what to do” are closely related to health behaviours. In the case of oral health, HL would affect a patient’s activities related to oral health such as regular tooth brushing and maintaining proper oral hygiene by using further aids such as interdental toothbrush or dental floss, etc. Lower health literacy would affect such behaviours, resulting in less frequent tooth brushing, using ineffective brushing techniques and not using further oral hygiene aids. Improper oral hygiene then results in the build-up of plaque, proliferating bacteria that cause inflammation or gingivitis. Although gingivitis is considered as a multifactorial disease associated with several risk factors and their combinations [26], mostly biological, genetic and environmental factors are considered in clinical practice. Our results show that health literacy is also an important factor in the path leading to gingivitis and shall be studied further in the causal pathways.

Long-term successfulness of multifactorial disease therapy can be achieved only by causally oriented treatment; that is, by eliminating individual changeable risk factors. It is also necessary to admit that such a disease cannot be definitively cured, especially if unchangeable risk factors are present. Success is considered when the disease is kept in a stable state, regularly motivating and encouraging the patient to cooperate and at the same time regularly checking his/her cooperation by repeated examinations [27]. Factors that can be changed by both the professional treatment and the patient include the oral microbiome, which is the basis for regular biofilm formation that damages the gingiva and the entire periodontium [28]. Unsuccessful therapies might be also caused by various shortcomings in dental practice. The requirements of customer-oriented dental practice include quality of service, respect and sensitivity. Clients appreciate the individual approach and friendly behaviour of all employees, seeing it as helpful and perceiving it positively [29]. The proper management of the dental clinic is based on proper communication [30]. Good communication is essential to the success of the whole practice. It serves as a means of management and mediates relationships between the staff and the patient. Communication can be a) one-way, where information spreads from the broadcaster to the recipient. It can also be a command-based two-way treatment where the attending and the client are involved in decisions and planning; b) multidirectional information exchange between multiple actors, e.g., doctor, patient, dental hygienist. Here, a uniform thinking of the whole team is important.

Nevertheless, unsuccessfulness of the dental therapy might also be related to patient HL level and by how professionals address this level and respond to it accordingly. Patients with a low HL level require more time, simpler language and a more structured therapeutic process compared to the patients with a high HL level. This approach is relatively new in dental offices but has become more common in other health care settings such as dialysis, oncology or cardiology where better results in health status could be achieved by addressing the health care needs of their patients and adjusting health care according to those needs for specific patients groups. Inclusion of patients’ HL level in dental health practices might increase successfulness of treatment efforts and improve oral health of the patients.

### Strengths and Limitations

We consider a strength of our study to be the analysis of a combination of objective data on oral health with subjective data on respondent’s health literacy, which is a weakness of other studies that provide just self-reported or clinical data. Another strength might be having collected a relatively large sample of adult respondents that covered almost all regions in Slovakia.

On the other hand, we are also aware of the limitations of the study. The cross-sectional design is one of the limitations, which does not allow us to make inferences about causality but only about associations. Ethical consideration in such a research environment would make an experimental study design not possible and that is why we preferred a cross-sectional study. Second, the sample was drawn from dental hygiene clinics, which does not allow us to generalize study findings on the entire Slovak adult population without constraint. It is possible that respondents attending oral hygiene appointments differ from the general population, although we do not have solid evidence about this. Third, we used only one objective clinical indicator of oral health status, despite the fact that there are several of them in use in dentist practices. We decided to use the CPITN because it was developed by the WHO as a universal index and it is used around the world, while other indexes are preferred in certain regions. As a last limitation, we used a relatively simple regression model with only the basic socioeconomic variables as adjustments. Further variables might be potential confounders, but assessing various confounders was not the aim of our particular study.

## 5. Conclusions

Our findings confirm that an individual’s lower health literacy is significantly associated with a higher chance of gingivitis incidence, which indicates problematic oral health related to improper oral hygiene. Health literacy might be a promising factor in the improvement of populations’ oral health, worthy of consideration in intervention and preventive activities.

## Figures and Tables

**Table 1 ijerph-17-02152-t001:** List of all nine health literacy questionnaire (HLQ) domains with a description of respondent characteristics with low and high levels of the particular domain.

Low Level of the Domain	High Level of the Domain
**HLQ 1. Feeling understood and supported by healthcare providers (4 items)**	
People who are low on this domain are unable to engage with doctors and other healthcare providers. They don’t have a regular healthcare provider and/or have difficulty trusting healthcare providers as a source of information and/or advice.	Has an established relationship with at least one healthcare provider who knows them well and who they trust to provide useful advice and information and to assist them to understand information and make decisions about their health.
**HLQ 2. Having sufficient information to manage my health (4 items)**	
Feels that there are many gaps in their knowledge and that they don’t have the information they need to live with and manage their health concerns.	Feels confident that they have all the information that they need to live with and manage their condition and to make decisions.
**HLQ 3. Actively managing my health (5 items)**	
People with low levels don’t see their health as their responsibility, they are not engaged in their healthcare and regard healthcare as something that is done to them.	Recognize the importance and are able to take responsibility for their own health. They proactively engage in their own care and make their own decisions about their health. They make health a priority.
**HLQ 4. Social support for health (5 items)**	
Completely alone and unsupported for health	A person’s social system provides them with all the support they want or need for health.
**HLQ 5. Appraisal of health information (5 items)**	
No matter how hard they try, they cannot understand most health information and get confused when there is conflicting information.	Able to identify good information and reliable sources of information. They can resolve conflicting information by themselves or with help from others.
**HLQ 6. Ability to actively engage with healthcare providers (5 items)**	
Are passive in their approach to healthcare, inactive i.e., they do not proactively seek or clarify information and advice and/or service options. They accept information without question. Unable to ask questions to get information or to clarify what they do not understand. They accept what is offered without seeking to ensure that it meets their needs. Feel unable to share concerns. The do not have a sense of agency in interactions with providers.	Is proactive about their health and feels in control in relationships with healthcare providers. Is able to seek advice from additional healthcare providers when necessary. They keep going until they get what they want. Empowered.
**HLQ 7. Navigating the healthcare system (6 items)**	
Unable to advocate on their own behalf and unable to find someone who can help them use the healthcare system to address their health needs. Do not look beyond obvious resources and have a limited understanding of what is available and what they are entitled to.	Able to find out about services and supports so they get all their needs met. Able to advocate on their own behalf at the system and service level.
**HLQ 8. Ability to find good health information (5 items)**	
Cannot access health information when required. Is dependent on others to offer information.	Is an ‘information explorer’. Actively uses a diverse range of sources to find information and is up to date.
**HLQ 9. Understanding health information well enough to know what to do (5 items)**	
Has problems understanding any written health information or instructions about treatments or medications. Unable to read or write well enough to complete medical forms.	Is able to understand all written information (including numerical information) in relation to their health and able to write appropriately on forms where required.

(source: Osborne et al. 2013).

**Table 2 ijerph-17-02152-t002:** Description of sociodemographic variables of the respondents and categories of patient with or without periodontal disease.

		Total		Gender				CPITN			
Variables		sample		Men		Women		Healthy		Periodontal disease	
*continuous variable*		M	*SD*	M	*SD*	M	*SD*	M	*SD*	M	*SD*
**Age**		36.4	*14.2*	35.8	*14.4*	36.7	*14.1*	34.5	*13.5*	43.8	*14.0*
	*t test value*	n/a		*T = –0.738*				*T = –7.729 ****			
*categorical variables*		N	%	N	%	N	%	N	%	N	%
**Gender**	men	402	*36.2*	--	--	--	--	289	35.4	59	38.8
	women	709	*63.8*	--	--	--	--	527	64.6	93	61.2
	*chi2 test value*	n/a		n/a				*chi2 = 0.643*			
**Education**	Primary education	33	*3.0*	12	*3.0*	21	*3.0*	24	*2.9*	5	*3.3*
	High school without graduation	108	*9.7*	44	*11.0*	64	*9.0*	66	*8.1*	20	*13.2*
	High school with graduation	580	*52.2*	200	*49.9*	378	*53.4*	434	*53.1*	83	*54.6*
	College/university	390	*35.1*	145	*36.2*	245	*34.6*	294	*35.9*	44	*28.9*
	*chi2 test value*	n/a		*chi2 = 1.766*				*chi2 = 5.656*			
**CPITN**	healthy (score 0)	818	*84.3*	289	*83.0*	527	*85.0*	--	--	--	--
	Periodontal disease (score 1 to 5)	152	*15.7*	59	*17.0*	93	*15.0*	--	--	--	--
	*chi2 test value*	n/a		*chi2 = 0.643*				n/a			

*** *p* < 0.001; -- not applicable; n/a = not available.

**Table 3 ijerph-17-02152-t003:** Difference in the level of health literacy domains between respondents with gingivitis and respondents with healthy teeth (t-test).

Health Literacy Domains	Respondents with Periodontal Disease		Respondents with Healthy Teeth		t-test Value
	M	*SD*	M	*SD*	
HLQ 1—Feeling understood and supported by healthcare providers	2.92	*0.49*	2.96	*0.48*	–0.873
HLQ 2—Having sufficient information to manage my health	2.69	*0.45*	2.83	*0.48*	–3.178 **
HLQ 3—Active health management	2.65	*0.50*	2.83	*0.49*	–3.904 ***
HLQ 4—Social support for health	03.5	*0.41*	03.1	*0.44*	–1.341
HLQ 5—Appraisal of health information	2.61	*0.54*	2.81	*0.49*	–4.367 ***
HLQ 6—Ability to actively engage with healthcare providers	3.33	*0.61*	3.51	*0.62*	–3.162 **
HLQ 7—Navigating the healthcare system	3.13	*0.60*	03.3	*0.62*	–2.97 **
HLQ 8—Ability to find good health information	3.35	*0.63*	03.6	*0.57*	–4.986 ***
HLQ 9—Understand health information well enough to know what to do	3.49	*0.54*	3.64	*0.55*	–3.029 **

** *p* < 0.01, *** *p* < 0.001.

**Table 4 ijerph-17-02152-t004:** Crude effects (odd ratios (OR) and 95% confidence interval (95% C.I.)) of health literacy domains and sociodemographic variables on the presence of periodontal disease, and health literacy domain effects on periodontal disease adjusted for sociodemographic variables.

HL Domain	Model Variables	Crude Effect on CPITN	Adjusted Effect on CPITN
		OR (95% CI)	OR (95% CI)
**HLQ1**	HL domain effect	0.85 (0.59–1.22)	0.82 (0.55–1.21)
	Age	1.05 (1.03–1.06) ***	1.05 (1.03–1.06) ***
	Gender (females vs. Males)	0.86 (0.61–1.23)	0.83 (0.57–1.20)
	Education	0.79 (0.63–0.99) *	0.86 (0.67–1.10)
**HLQ2**	HL domain effect	0.55 (0.38–0.79) ***	0.59 (0.40–0.88) **
	Age	--^a^	1.05 (1.0302–1.06) ***
	Gender (females vs. Males)	--^b^	0.81 (0.56–1.19)
	Education	--^c^	0.89 (0.70–1.15)
**HLQ3**	HL domain effect	0.50 (0.35–0.71) ***	0.54 (0.37–0.78) ***
	Age	--^a^	1.05 (1.03–1.06) ***
	Gender (females vs. Males)	--^b^	0.846 (0.57–1.22)
	Education	--^c^	0.90 (0.699–1.15)
**HLQ4**	HL domain effect	0.7600 (0.51–1.14)	0.84 (0.55–1.29)
	Age	--^a^	1.05 (1.03–1.06) ***
	Gender (females vs. Males)	--^b^	0.81 (0.56–1.18)
	Education	--^c^	0.84 (0.66–1.08)
**HLQ5**	HL domain effect	0.46 (0.32–0.66)	0.55 (0.38–0.80) **
	Age	--^a^	1.04 (1.030–1.056) ***
	Gender (females vs. Males)	--^b^	0.86 (0.59–1.25)
	Education	--^c^	0.93 (0.72–1.19)
**HLQ6**	HL domain effect	0.64 (0.49–0.85) **	0.65 (0.49–0.87) **
	Age	--^a^	1.05 (1.03–1.06) ***
	Gender (females vs. Males)	--^b^	0.85 (0.58–1.24)
	Education	--^c^	0.90 (0.71–1.16)
**HLQ7**	HL domain effect	0.65 (0.49–0.86) **	0.66 (0.49–0.88) **
	Age	--^a^	1.05 (1.03–1.06) ***
	Gender (females vs. Males)	--^b^	0.79 (0.54–1.15)
	Education	--^c^	0.89 (0.69–1.13)
**HLQ8**	HL domain effect	0.48 (0.354–0.64) ***	0.58 (0.42–0.79) ***
	Age	--^a^	1.04 (1.03–1.06) ***
	Gender (females vs. Males)	--^b^	0.83 (0.57–1.22)
	Education	--^c^	0.90 (0.70–1.16)
**HLQ9**	HL domain effect	0.62 (0.45–0.85) **	0.63 (0.45–0.88) **
	Age	--^a^	1.05 (1.03–1.06) ***
	Gender (females vs. Males)	--^b^	0.85 (0.58–1.24)
	Education	--^c^	0.92 (0.72–1.18)

** *p* < 0.01, *** *p* < 0.001; -- not applicable; ^a^ crude effect of the age is the same as reported in the HLQ1 box; ^b^ crude effect of the gender is the same as reported in the HLQ1 box; ^c^ crude effect of the education is the same as reported in the HLQ1 box.

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
