# Peer review of "Health Literacy Associations with Periodontal Disease among Slovak Adults"

_ijerph, 2020, doi:10.3390/ijerph17062152_

Round 1

Reviewer 1 Report

The manuscript entitled "Health literacy associations with periodontal disease among Slovak adults" has been reviewed. Authors presented the results of a cross-sectional study, designed to examine the associations of Health Literacy and an indicator of gingivitis among Slovak adults. It has a good scientific background and study design. Below are some suggestions to improve the quality of manuscript:

  1. Introduction section, lines 67-80: The mechanism of gingivitis and its association with other disease seems irrelevant to the scope of this study.

  1. Materials and Methods: Authors used valid and reliable measures, however, there is no information about the language of questionnaires. Was it in Slovak language or English? If Slovak, they need to provide more information about the validity and reliability of the Slovak version of these manuscripts. If they used English version, the English literacy of participants should be presented.

  1. Results, Table 3: Why the crude effects of age, education, and gender on CPITN is just presented for HLQ1 not other items? Please explain.

  1. Discussion: Authors did a great job in explaining the limitations of this study. Considering these limitations, any interpretation should be limited to the Slovak adults, not the whole general population. It should be carefully considered in the concluding statement in the abstract and conclusion section.

Author Response

We would like to express our gratitude to all three reviewers who thoroughly assessed our manuscript and provided us with important comment that helped us to improve our manuscript further. We hope that we solved all raised issues. Our detailed response how we responded to each comment is below in point-by-point manner.

Reviewer 1

Comments and Suggestions for Authors

The manuscript entitled "Health literacy associations with periodontal disease among Slovak adults" has been reviewed. Authors presented the results of a cross-sectional study, designed to examine the associations of Health Literacy and an indicator of gingivitis among Slovak adults. It has a good scientific background and study design. Below are some suggestions to improve the quality of manuscript:

Reviewer’s Comment 1.1: Introduction section, lines 67-80: The mechanism of gingivitis and its association with other disease seems irrelevant to the scope of this study.

Reply 1.1: Thank you for the substantial comment. We agree with you and we removed the description of the gingivitis mechanism.

Adjustment in the text 1.1: Paragraph describing the gingivitis mechanism was removed from introduction.

Reviewer’s Comment 1.2: Materials and Methods: Authors used valid and reliable measures, however, there is no information about the language of questionnaires. Was it in Slovak language or English? If Slovak, they need to provide more information about the validity and reliability of the Slovak version of these manuscripts. If they used English version, the English literacy of participants should be presented.

Reply 1.2: We agree that language of the questionnaire is important to mention and information about validity of the HLQ should be mentioned. We thought that providing the reference to the study which reports translation and validation process of the Slovak version of the HLQ, but providing more detailed information about the questionnaire would be more desirable.

Adjustment in the text 1.2: Detailed information about HLQ-SK validation to the 2.2. Measures paragraph was added:

The battery of questioners contained various items and scales covering different variables related to respondents’ oral health, and health related behaviours. For the purposes of this paper, we used demographics (e.g. gender, age), socioeconomic status (educational attainment) and Health literacy questionnaire (HLQ-SK). Questionnaire battery was in Slovak language and originally English questionnaires were used in Slovak version, that was officially translated and validated. Objective oral health status was evaluated using Community Periodontal Index of Treatment Needs (CPITN).

“Translation, adaptation and validation of the questionnaire followed specific translation procedure developed by the HLQ authors. HLQ-SK replicated factor structure of the English HLQ factor structure (satisfactory goodness of fit [χ2WLSMV = 1684.96 (df = 866), p < 0.001; CFI = 0.943, TLI = 0.938, RMSEA = 0.051, and WRMR = 1.297] and achieved acceptable internal consistency and component reliability - Cronbach’s alphas and composite reliability coefficients ranged from 0.73 to 0.84 [26]. The original HLQ is divided into two parts which differ in response categories. Part 1 (domains 1–5) has 4 response categories rating the extent of agreement (see Table 1). Part 2 (domains 6–9) has 5 response categories rating the level of difficulty: cannot do or always difficult (1), usually difficult (2), sometimes difficult (3), usually easy (4) and always easy (5). Each domain was scored as the average of the item scores.”

Reviewer’s Comment 1.3: Results, Table 3: Why the crude effects of age, education, and gender on CPITN is just presented for HLQ1 not other items? Please explain.

Reply 1.3: Thank for the comment. Probably you wanted to mention Table 4. Table 4 presents effect of crude and adjusted effect of the variables on the CPITN score. Crude effect of age, education, and gender are tested individually and presented only in the first block of the table because there are still same. But because we wanted to show change in crude and adjusted effect of HLQ-SK domains we wanted to present them next to each other but crude effect do not change for the same age, education, and gender variables.

Adjustment in the text 1.3: explanation in the table footnote was added

a crude effect of the age is the same as reported in the HLQ1 box

b crude effect of the gender is the same as reported in the HLQ1 box

c crude effect of the education is the same as reported in the HLQ1 box

Reviewer’s Comment 1.4: Discussion: Authors did a great job in explaining the limitations of this study. Considering these limitations, any interpretation should be limited to the Slovak adults, not the whole general population. It should be carefully considered in the concluding statement in the abstract and conclusion section.

Reply 1.4: We agree with your comments. We tried to limit our conclusion just for Slovak adults attending oral hygiene clinics.

Adjustment in the text 1.4: Concluding sentence in Abstract was extended as follows:

Our findings confirm that individual’s lower HL is significantly associated with higher chance of periodontal disease incidence, specifically among Slovak adults attending oral hygiene clinics.

Reviewer 2 Report

In this manuscript, the authors investigated the effect of health literacy on periodontal disease in Slovak adults. The question was straightforward and the experimental approach was simple: they assessed the literacy level through questionnaire, and correlated with incidence and severity of periodontal diseases.

The major issue that needs to be addressed is the lack of description of key methodological approaches and study design, which makes it difficult to judge the scientific merits of this study. For example, it is unclear how the Health Literacy Questionnaire (HLQ) works. Methodological description is needed: general readers need to be made aware of the experimental approach without referring to background literature. Were the questions narrative and then quantified by questionnaire administers? How were qualitative data converted into quantitative parameters? What are the scales of value for quantification? Who did the quantitation and were they blinded?

Who were the subjects, and how were they recruited and selected?

The table of questionnaire used quite subjective terms, such as “feels”, and vague adjectives. How do these terms translate into objective quantitative parameters?

How does Community Periodontal Index of Treatment Needs (CPITN) work? Brief explanation is needed in the Methods section rather than merely quoting for a reference. Who assessed CPITN?

Author Response

We would like to express our gratitude to all three reviewers who thoroughly assessed our manuscript and provided us with important comment that helped us to improve our manuscript further. We hope that we solved all raised issues. Our detailed response how we responded to each comment is below in point-by-point manner.

Reviewer 2

Comments and Suggestions for Authors

In this manuscript, the authors investigated the effect of health literacy on periodontal disease in Slovak adults. The question was straightforward and the experimental approach was simple: they assessed the literacy level through questionnaire, and correlated with incidence and severity of periodontal diseases.

Reviewer’s Comment 2.1: The major issue that needs to be addressed is the lack of description of key methodological approaches and study design, which makes it difficult to judge the scientific merits of this study. For example, it is unclear how the Health Literacy Questionnaire (HLQ) works. Methodological description is needed: general readers need to be made aware of the experimental approach without referring to background literature. Were the questions narrative and then quantified by questionnaire administers? How were qualitative data converted into quantitative parameters? What are the scales of value for quantification? Who did the quantitation and were they blinded?

Reply 2.1: Thank you for pointing out our unclear description of our study design and especially HLQ. We checked the design description and added substantial information to the text. Except already mentioned: cross-sectional study design, combined data collection using self-administered questionnaire and objective oral health data, We emphasised that both types of data were collected by dental hygienist that were trained for that task and we described the how patients were invited to participate in the study alongside with actual response rate achieved. Detailed information about HLQ were also added. HLQ is not experimental questionnaire. It contains 44 items divided into 2 parts and those 44 items constitutes 9 domains of health literacy. Respondents expresses the level of agreement with the statements in the first part (domains 1–5) - 4 response categories rating the extent of agreement: strongly disagree (1), disagree (2), agree (3), strongly agree (4) (see Table 1). Part 2 (domains 6–9) has 5 response categories rating the level of difficulty: cannot do or always difficult (1), usually difficult (2), sometimes difficult (3), usually easy (4) and always easy (5). Each domain was scored as the average of the item scores

Adjustment in the text 2.1: We expanded the description of the HLQ-SK in the Methods section:

Health literacy was measured using Slovak version of Health literacy questionnaire (HLQ-SK) [22,23] which consisted of 44 items divided into 9 subscales of health literacy (see Table 1). Translation, adaptation and validation of the questionnaire followed specific translation procedure developed by the HLQ authors. HLQ-SK replicated factor structure of the English HLQ factor structure (satisfactory goodness of fit [χ2WLSMV=1684.96 (df=866), p < 0.001; CFI=0.943, TLI=0.938, RMSEA=0.051, and WRMR=1.297] and achieved acceptable internal consistency and component reliability - Cronbach’s alphas and composite reliability coefficients ranged from 0.73 to 0.84 [22]. The original HLQ is divided into two parts which differ in response categories. Part 1 (domains 1–5) has 4 response categories rating the extent of agreement (see Table 1). Part 2 (domains 6–9) has 5 response categories rating the level of difficulty: cannot do or always difficult (1), usually difficult (2), sometimes difficult (3), usually easy (4) and always easy (5). Each domain was scored as the average of the item scores [23]. Higher score indicates higher level of HL abilities in particular domain.

Reviewer’s Comment 2.2: Who were the subjects, and how were they recruited and selected?

Reply 2.2: We provided explanation that you request by adding following text into the Methods section.

Adjustment in the text 2.2:

Patients attending dental hygiene treatment were asked to participate in the study. Brief explanation of the study rationale and informed consent form were provided by the hygienist before treatment. Patients who agreed to participate filled the questionnaire after dental hygiene treatment. From the 1479 invited patients 75,5% participated in the study.

Reviewer’s Comment 2.3: The table of questionnaire used quite subjective terms, such as “feels”, and vague adjectives. How do these terms translate into objective quantitative parameters?

Reply 2.3:  Table 1 was presented in the paper to provide better understanding of the health literacy components – 9 subscales of the HLQ. The table describes patients with extremely low or extremely high levels of particular subscale. Each subscale level result from respondent’s answers on 4-6 items per subscale in the questionnaire. Items are formulated as a statements and respondent has to express the degree of agreement (in first part) and in second part of the questionnaire the respondent has to rate the level of difficulty of the proposed action in the item. In this way the mechanism of the questionnaire is not different from others questionnaires used in research.

Regarding the “subjective terms” that are used to describe the content of the health literacy domains we have to admit that you are right. Unfortunatelly we are not able to provide less subjective or more objective terms. On the other hand we used the same characteristics as were provided by the original author. We would also like to mention that subjective terms reflects the intended content of the scales which aims to capture those specific characteristics, experiences and abilities of the respondents. Questionnaires or scales were specifically designed by psychologist to measure, capture subjective qualities and experiences that are not measurable or observable by objective means. Thus certain subjectivity of the concepts measured by questionnaires, even HLQ, is inherent characteristic of the method. The way of constructing and using such questionnaires is covered by psychometrics - a field of study concerned with the theory and technique of psychological measurement and are for a decades accepted as a valid method for measuring specific variables (e.g. quality of life, depression, life satisfaction, pain). Development of HLQ and its translation into Slovak language followed strict psychometric standards and met all required levels of model fit indices, internal structure and internal reliability indicators.

Adjustment in the text 2.3:

Table 1 heading was reformulated to better explain the table content.

“List of all nine Health literacy questionnaire (HLQ) domains with description of respondent characteristic with low and high level of the particular domain”

We also indicated number of items by each domain title.

Reviewer’s Comment 2.4: How does Community Periodontal Index of Treatment Needs (CPITN) work? Brief explanation is needed in the Methods section rather than merely quoting for a reference. Who assessed CPITN?

Reply 2.4: We agree that our explanation of CPITN was not sufficient. We have formulated broader explanation, which is provided below

Adjustment in the text 2.4: We have expanded the test in Methods section with following text

Oral health status (presence of periodontal disease) was expressed as a specific standardized index called Community Periodontal Index of Treatment Needs (CPITN) [27]. CPITN was originally designed for describing periodontal treatment needs in populations. It has also been used to describe the prevalence of periodontal conditions and as a screening test to identify patients who need complex or simple treatment [25]. CPITN records the common treatable conditions – periodontal, pockets, gingival inflammation (identified by bleeding on a gentle probing), and dental calculus and other plaque retentive factors. It does not record irreversible changes such as recession or other deviation from periodontal health such as tooth mobility or loss of periodontal attachment. The dentition is divided into six parts (sextants: 17-14, 13-23, 24-27, 37-34, 33-43, 44-47) and each sextant is given a score. Highest score in each sextant is identified after examining all teeth: no need for care (score 0), bleeding gingivae on gentle probing (score 1), presence of calculus and other plaque retentive factors (score 2), and presence of 4 or 5 mm pockets (score 3) or 6 or deeper pockets (score 4). Use of a special 621 WHO periodontal probe (or its equivalent) is recommended [11]. We used CPITN as dichotomized indicator of healthy teeth and teeth with periodontal disease (dichotomized as: 0 – healthy teeth, 1 – periodontal disease).

Reviewer 3 Report

Thank you for the opportunity to contribute to the peer review process for the original study submission manuscript entitled “Health literacy associations with periodontal disease among Slovak adults”. It is a very interesting and relevant study that found an association with most of the domains of heathy literacy with periodontal disease.

However, I indicate below some minor revisions to be done.

Abstract

Line 27 – Please amend the OR value

Introduction

Line 77 – C-reactive protein (CRP), it appears again in line 80 just as CRP.

Material and Methods

There is some repeated information on sections 2.1 and 2.2. You could describe the “clinics across Slovakia” more detailed. How were they selected? How many different clinics? As it is an international journal, the readers do not know about health care system in Slovakia, so it is important to describe if these clinics are public or private, the estimated population attended by these clinics, etc.

Line 101 – adults are individuals 18y or older or 21y or older? Please specify.

The definition of CPITN appears just on line 121, although it is described in your objectives. I suggest a short explanation in introduction, including why this measure was chosen based on literature.

Line 126 – I suggest putting the information “(dichotomized as….)” as a real phrase, not just a parenthesis.

Results

Line 134 – correct “descriptive statics

Line 138 – “because of insignificant 138 association with CPITN.” The correct statistical term is “non-significant”, but you have to specify the significant level (0,05?).

Table 2 – format the table properly.

              The Healthy category is closer than Gender instead of CPITN

              Add “.0” for % to facilitate reading

Line 150 – Linear or Logistic?

Discussion

I think the sentence in line 171 cannot be stated as something found in your study because you did not measure oral hygiene of the participants, and as you stated in line 190: “Although gingivitis is considered as a multifactorial disease caused by several risk factors and their combinations”, so we can infer that “the higher chance of gingivitis prevalence (not incidence) indicates problematic oral health”, but cannot state that this is “caused by improper oral hygiene” as you did not measure this (or other) risk factors in your sample.

Line 171 – “We found that respondent‘s lower health literacy is significantly associated with higher chance of gingivitis incidence what indicates problematic oral health related caused by improper oral hygiene

Line 190 – “Although gingivitis is considered as a multifactorial disease caused by several risk factors and their combinations”

Line 187 – Tooth brushing

Line 193 – Shall

Author Response

We would like to express our gratitude to all three reviewers who thoroughly assessed our manuscript and provided us with important comment that helped us to improve our manuscript further. We hope that we solved all raised issues. Our detailed response how we responded to each comment is below in point-by-point manner.

Reviewer 3

Comments and Suggestions for Authors

Thank you for the opportunity to contribute to the peer review process for the original study submission manuscript entitled “Health literacy associations with periodontal disease among Slovak adults”. It is a very interesting and relevant study that found an association with most of the domains of heathy literacy with periodontal disease.

However, I indicate below some minor revisions to be done.

Reviewer’s Comment 3.1: Abstract, Line 27 – Please amend the OR value

Reply 3.1: Thank you for spotting that OR was not provided accordingly. We removed dash by mistake. We have corrected that

Adjustment in the text 3.1:

(adjusted ORs 0.55-0.67, p<0.05)

Reviewer’s Comment 3.2: Introduction, Line 77 – C-reactive protein (CRP), it appears again in line 80 just as CRP.

Reply 3.2: We removed whole paragraph describing the inflammation process as it was suggested by the reviewer 1.

Adjustment in the text 3.2: none

Reviewer’s Comment 3.3: Material and Methods, There is some repeated information on sections 2.1 and 2.2. You could describe the “clinics across Slovakia” more detailed. How were they selected? How many different clinics? As it is an international journal, the readers do not know about health care system in Slovakia, so it is important to describe if these clinics are public or private, the estimated population attended by these clinics, etc.

Reply 3.3: In Slovakia we do not have an official network of dental hygiene clinics which we may refer to as particular number of such clinics. Dental hygiene is practised as private practice and dental hygiene clinic are opened within dentist offices or practised as individual dental hygiene practices. Number of such practices is not officially available although the number of dental hygiene clinics and practices is increasing in recent time. For the purposes of our study we addressed all dental hygiene clinics within dentist offices (N=35) who collaborate with Department of Dental Hygiene (Faculty of Health Care, University of Prešov) and 1st Department of Stomatology (Faculty of Medicine, P.J. Šafarik University). Department of Dental Hygiene provided training and supervision and 1st Department of Stomatology provided consultations for dental hygienists in those clinics. In such way dental hygiene clinics which agreed to participate (N=28) in the study get further training for questionnaire administration and oral health assessment in the standardized way. Dental hygiene clinics participating in the study are situated around whole Slovakia, almost evenly. Because of nonexistence of official and complete list of all dental health practices we were not able to use random sampling of such offices to assure better methodological cleanliness. Despite we use as a population of dental health clinics database of our “partners” and we asked all of them to participate.

Adjustment in the text 3.3:

Participating dental hygiene clinics were recruited from the collaborating dental hygiene clinics (N=35) where Department of Dental Hygiene (Faculty of Health Care, University of Prešov) and 1st Department of Stomatology (Faculty of Medicine, P.J. Šafarik University) provided training, supervision or consultations for dental hygienists in those clinics. All dental hygiene clinics were private clinics what reflects situation in Slovakia.

Reviewer’s Comment 3.4: Line 101 – adults are individuals 18y or older or 21y or older? Please specify.

Reply 3.4: We consider individuals older than 18 years as adults.

Adjustment in the text 3.4: We added this information into the text

Patients attending dental hygiene treatment older than 18 years were asked to participate in the study.

Reviewer’s Comment 3.5: The definition of CPITN appears just on line 121, although it is described in your objectives. I suggest a short explanation in introduction, including why this measure was chosen based on literature.

Reply 3.5: We agree that some kind of justification of the CPITN selection should be provided. There are several indexes used for assessment of periodontal health such as Periodontal disease index (PDI), CPITN and Periodontal screening and recording (PSR) and their modification such as Basic periodontal examination (BPE). There are also other indexes focusing on specific issues in oral health: Plaque index, API – Approximal plaque index, SBI – Sulcus bleeding index, PBI – papilla bleeding index, BOP – bleeding on probing, etc.

Our selection of CPITN was based on the fact that the index was originally intended to be used as screening and epidemiological tool which is easy to use and providing valid results in larger samples (Cutress, T.W 1987). CPITN is also largely used in epidemiological research as periodontal health and treatment needs indicator. Furthermore, the CPITN fits the best the purpose of our study – to analyse the association of HL with periodontal disease, which is according to our opinion suitably assessed by this index.

Adjustment in the text 3.5: We added following paragraph into the Introduction of our paper.

There are several indexes used for assessment of periodontal health such as Periodontal disease index (PDI), Community periodontal index of treatment needs (CPITN) and Periodontal screening and recording (PSR) (Wolf et al 2005) and their modification such as Basic periodontal examination (BPE) (Gjermo 1994). CPITN was developed by WHO to provide valid screening and epidemiological tool for quick assessment of periodontal health in larger samples and easy comparison among populations (Cutress 1987).

Reviewer’s Comment 3.6: Line 126 – I suggest putting the information “(dichotomized as….)” as a real phrase, not just a parenthesis.

Reply 3.6: We agree with your proposal and we formulated sentence about dichotomisation.

Adjustment in the text 3.6: Sentence about dichotomisation was added into the description of CPITN

…We used CPITN as dichotomized indicator of healthy teeth (score 0 – coded as 0) and teeth with periodontal disease (scores 1-4 – coded as 1).

Reviewer’s Comment 3.7: Results, Line 134 – correct “descriptive statics

Reply 3.7: Thank you for spotting this typo.

Adjustment in the text 3.7: Corrected

Firstly, descriptive statistics was…

Reviewer’s Comment 3.8: Line 138 – “because of insignificant 138 association with CPITN.” The correct statistical term is “non-significant”, but you have to specify the significant level (0,05?).

Reply 3.8: Thank you for this comment. We have corrected the text and added the notion about the significance level.

Adjustment in the text 3.8:

…because of non-significant association with CPITN at the level of p<0.05.

Reviewer’s Comment 3.9: Table 2 – format the table properly. The Healthy category is closer than Gender instead of CPITN, Add “.0” for % to facilitate reading

Reply 3.9: Thank you for raising this issue. We have formatted the table evenly and added, missing zeros in percentage numbers. We also used italics for percentages to better distinguish this from counts.

Adjustment in the text 3.9: Table was formatted, zeros added

Reviewer’s Comment 3.10: Line 150 – Linear or Logistic?

Reply 3.10: We used logistic regression but by mistake we referred to linear regression in Results section. We corrected it into “logistic” regression.

Adjustment in the text 3.10: linear changed into logistic

Further, the results of logistic regression analyses confirmed

Reviewer’s Comment 3.11: Discussion, I think the sentence in line 171 cannot be stated as something found in your study because you did not measure oral hygiene of the participants, and as you stated in line 190: “Although gingivitis is considered as a multifactorial disease caused by several risk factors and their combinations”, so we can infer that “the higher chance of gingivitis prevalence (not incidence) indicates problematic oral health”, but cannot state that this is “caused by improper oral hygiene” as you did not measure this (or other) risk factors in your sample.

Reply 3.11: We agree with your comment and we tried to rephrase the statements in the way to soften the strong wording.

Adjustment in the text 3.11: see below

Reviewer’s Comment 3.12: Line 171 – “We found that respondent‘s lower health literacy is significantly associated with higher chance of gingivitis incidence what indicates problematic oral health related caused by improper oral hygiene

Reply 3.12: We have removed the word “caused” which is redundant in that sentence

Adjustment in the text 3.12:

We found that respondent‘s lower health literacy is significantly associated with higher chance of gingivitis prevalence what indicates problematic oral health usually related to improper oral hygiene.

Reviewer’s Comment 3.13: Line 190 – “Although gingivitis is considered as a multifactorial disease caused by several risk factors and their combinations”

Reply 3.13:

Adjustment in the text 3.13:

Although gingivitis is considered as a multifactorial disease associated with several risk factors and their combinations [30].

Reviewer’s Comment 3.14: Line 187 – Tooth brushing

Reply 3.14: Thank you for spotting this typo.

Adjustment in the text 3.14: We have corrected it

…less frequent tooth brushing,

Reviewer’s Comment 3.15: Line 193 – Shall

Reply 3.15: Thank you for spotting this typo.

Adjustment in the text 3.15: We have corrected it

…leading to gingivitis and shall be studied

Round 2

Reviewer 2 Report

The authors have addressed my previous concerns. Writing needs to be significantly improved for publication.